# Artificial Neurons on Flexible Substrates: A Fully Printed Approach for Neuromorphic Sensing

**DOI:** 10.3390/s22114000

**Published:** 2022-05-25

**Authors:** Surya A. Singaraju, Dennis D. Weller, Thurid S. Gspann, Jasmin Aghassi-Hagmann, Mehdi B. Tahoori

**Affiliations:** 1Institute of Nanotechnology, Karlsruhe Institute of Technology, Hermann-von-Helmholtz-Platz 1, 76344 Eggenstein-Leopoldshafen, Germany; dennis.weller@kit.edu (D.D.W.); thurid.gspann@kit.edu (T.S.G.); 2Institute of Computer Science & Engineering, Karlsruhe Institute of Technology, Kaiserstrasse 12, 76131 Karlsruhe, Germany; tahoori@ira.uka.de

**Keywords:** printed electronics, flexible and functional inks, artificial neural networks, neuromorphic sensing and computing

## Abstract

Printed electronic devices have demonstrated their applicability in complex electronic circuits. There is recent progress in the realization of neuromorphic computing systems (NCSs) to implement basic synaptic functions using solution-processed materials. However, a fully printed neuron is yet to be realised. We demonstrate a fully printed artificial neuromorphic circuit on flexible polyimide (PI) substrate. Characteristic features of individual components of the printed system were guided by the software training of the NCS. The printing process employs graphene ink for passive structures and In${}_{2}$O${}_{3}$ as active material to print a two-input artificial neuron on PI. To ensure a small area footprint, the thickness of graphene film is tuned to target a resistance and to obtain conductors or resistors. The sheet resistance of the graphene film annealed at 300 °C can be adjusted between 200 Ω and 500 kΩ depending on the number of printed layers. The fully printed devices withstand a minimum of 2% tensile strain for at least 200 cycles of applied stress without any crack formation. The area usage of the printed two-input neuron is 16.25 mm${}^{2}$, with a power consumption of 37.7 mW, a propagation delay of 1 s, and a voltage supply of 2 V, which renders the device a promising candidate for future applications in smart wearable sensors.

## 1. Introduction

### 1.1. Printed Electronics in Neuromorphic Sensing

Printed electronics (PE) are an emerging fabrication technology that5 enable new innovative products and application fields to drive the ever-increasing intelligent sensor market through smart front-end devices. It can substantially decrease manufacturing costs and the time to market, and allows for hardware with a flexible form factor. New application domains that require low-cost flexible electronics are targeted, since conventional silicon-based technologies possess form-factor limitations [1,2,3]. Typical applications are soft sensors [4,5], soft robotics [6,7,8], flexible wearable medical devices [9], fast-moving smart consumer goods [2,10,11], or Internet of Things (IoT) infrastructures [12]. Similar to conventional silicon technology, sensor preconditioning and processing can also be performed for printed technology [13], preferably with the deployment of power- and area-efficient neuromorphic computing systems (NCSs). As an example, NCS can be used to complement highly sensitive soft robotic pressure sensors [14] for optimizing the grabbing and releasing of objects.

Printed electronics rely on digital additive processes such as inkjet printing, and could be printed directly on flexible materials without requiring a mask or subtractive processes [15,16,17,18]. Due to the difficulty in printing dielectric materials, we chose an electrolyte gating approach that was inkjet-printer-friendly. Therefore, electrolyte-gated transistor technology (EGT) [19] is a promising candidate. It can be fully processed with digital drop-on-demand properties systems without the need for additional postprocessing steps, and thus fulfils the required for flexible, point-of-use, and low-cost electronics. Furthermore, EGTs can be driven by very low supply voltages, as low as 2 V, allowing for using printed batteries or energy harvesters for the power supply for portable applications. Owing to these offered benefits by EGTs, there are significant efforts in utilizing them for neuromorphic sensing, synaptic coupling, and artificial sensory systems [20].

Conventional electric circuit designs cannot be directly mapped to our EGT technology owing to missing PMOS, larger feature size and lower integration density. Even though there is progress in organic electronics [21], a PMOS is still missing in printed inorganic electronics and is not yet optimized for CMOS. Hence digital systems in the technology lead to substantial area overhead, high power consumption and low performance. Consequently, the requirements of the targeted applications cannot be met [22]. In that respect, biology-inspired neuromorphic computing [23] coupled with analog computing [24,25] can be leveraged to obtain an efficient circuit implementation for PE applications. An NCS is inspired by the topology of the brain and can be built from artificial synapses and neurons. A significant advantage of the NCS is that sensory data can be processed without having to convert into digital signals, which otherwise requires additional analog–digital converters which increase costs and the hardware footprint. The intrinsic fault tolerance of NCS [26,27] can be leveraged to tolerate manufacturing imperfections and defects, which is a benefit for printed systems. The challenge, however, is to fully print the neuromorphic circuits.

### 1.2. Device Architecture of Fully Printed Two-Input Neuron

In general, very common realizations of NCS are based on feed-forward or artificial neural networks (ANNs) [28], which are adapted in training routines such as back-propagation [29]. Printed NCSs consist of a network of artificial neurons comprising a network of resistors that are both intra- and interlayer interconnected to perform regression or classification tasks [29]. ANNs are implemented into the hardware owing to their simplicity and ease of training, which can be achieved by a least-mean-squares model. Most recently, artificial neural networks fabricated by additive technologies were presented, such as basic synaptic functions on flexible substrates [30] and a multiply–accumulate (MAC) operation on flexible substrates was also presented [31]. NCSs based on EGTs [32,33,34], and screen-printed [35] and aerosol-jet-printed NCS [36] were reported. Some works proposed organic functional materials [24] and suitable training algorithms for printed NCSs [33,34,37,38]. Although these contributions strongly indicated the relevance of printed NCS for prospective computing systems, most of the works investigating EGT-based NCSs still use rigid substrates such as silicon and glass. Moreover, passive conductive tracks are often not printed but thermally evaporated or sputtered, and hence require additional subtractive processes such as photolithography or laser ablation. Recent works [34] that reported printed NCS utilized laser ablation to fabricate resistors that encompass a large area overhead owing to the requirement of large resistances. Instead, a fully printed NCS system that includes printed resistors would greatly reduce the resistor area. However, only few works exist on devices fully printed on flexible substrates, and if so, are mainly screen printed [30].

In this article and, to our knowledge, for the first time, we present a fully printed and flexible two-input neuron for NCSs based on low-voltage EGT technology. The main advantage of the proposed neuron design and its fabrication process is its capability to be embedded as near-sensor processing units for preprocessing raw sensor data, such as matrix of pressure sensors as part of electronic skin. The NCS is first trained in software, and the proposed design is fabricated using easy additive-printing steps.

We circumvented previously deployed nonadditive structuring processes and substituted them with additive-printing processes for the EGT-based NCS on flexible polyimide (PI). We used graphene ink for conductive lines to act as resistors or wires depending on their shape, and utilized In${}_{2}$O${}_{3}$ as the active channel material in the EGT owing to its superior performance on flexible substrates such as PI [39].

### 1.3. Proposed Two-Input Neuron

As illustrated in Figure 1, by replicating the neuron intra- and interlayer-wise, a full ANN can be formed for a given topology, which is similar to conventional (software-based) ANNs with few additional constraints that would need to be considered, such as bounded and coupled ANN weights. Thus, each layer of the ANN that is trained in software is fully printed. Given the ease of fabrication and the reproducibility of printed ANNs, a multiple-input neuron can be easily designed and printed with the proposed design.

## 2. Materials and Methods

### 2.1. Ink Preparation and Characterization

In this work, an inkjet printer (Dimatix 2831, Fujifilm, Santa Clara, CA, USA) and a Microplotter II (Sonoplot, Middleton, WI, USA) were utilized in order to fully print the two-input neuron. The inkjet printer had a fixed nozzle diameter of 20 μm, while the nozzle diameter in the Microplotter could be increased to for the easier printability of the graphene ink [40]. This removes the need for filtering large flakes out beforehand.

The In${}_{2}$O${}_{3}$ semiconductor was prepared with a a precursor method, namely, 0.05 M $\mathrm{In}{\left(NO_{3}\right)}_{3}$ dissolved in a 4:1 water:ethylene glycol mixture was inkjet-printed and annealed at 350 °C. The precursor ink was stirred at 300 rpm for 12 h to ensure homogeneous mixing before filtering it through a PVDF filter and printing. For passive graphene structures, commercial ink (2.4 wt % solid graphene and ethyl cellulose in cyclohexanone and terpineol, Sigma Aldrich, Taufkirchen, Germany) was used. The graphene ink was printed with the Microplotter by adjusting the printer nozzle diameter to 30 μm. The inkjet-printable composite solid polymer electrolyte (CSPE) was prepared by mixing two solutions. The first solution was a base solvent comprising 0.63 g of polyvinyl alcohol (PVA, hydrolyzed, 98%, Sigma-Aldrich) dissolved in 6 g dimethyl sulphoxide (DMSO, anhydrous, 99.9%, Sigma-Aldrich), stirred at 300 rpm for 1 h at 85 °C. The second solution was the electrolyte, which was prepared by dissolving 0.07 g ${\mathrm{LiClO}}_{4}$ (anhydrous, 98%, Alpha Aesar, Kandel, Germany) in 3 g of propylene carbonate (PC, anhydrous, 99.7%, Sigma-Aldrich) and stirring at 300 rpm for 1 h. The two solutions were then stirred together at 300 rpm overnight before printing.

The material stack of the printed inplane electrolyte gated transistor (EGT), which was the basis for the two-input neuron, is depicted in Figure 2. The cross-section of the EGT is shown in Figure 2b with the graphene source/drain, the In${}_{2}$O${}_{3}$ active layer, and the composite solid polymer electrolyte (CSPE) that performed the role of a dielectric in the printed transistor. A channel was formed on the In${}_{2}$O${}_{3}$ semiconductor surface in contact with the CSPE due to the formation of a double electrical layer that attracted charge carriers onto the surface [41]. EGT channel length was 200 μm, width was 100 μm, the S/D was about 300 × 300 μm, whereas the inplane gate electrode was 750 × 500 μm. The overall size of the transistor was thus 1 × 1.5 mm. A scanning electron microscope image of the top view of the device, and the interface between graphene and In${}_{2}$O${}_{3}$ are shown in Figure 2c.

### 2.2. Electrical Analysis of Printed Two-Input Neuron

Electrical analysis was conducted with the Agilent 156C (Keysight, Böblingen, Germany) Semiconductor Parameter Analyzer in combination with the Yokogawa DL6104 digital oscilloscope (Wehr, Germany) and Keithley 3390 waveform generators (Tektronix, Köln, Germany) for input pulse generation. In total, two types of measurements were performed: DC characterization of the EGT-based diode (gate and drain of transistor $T_{1}$ shortcut by $R_{H}$), and transient simulation of the whole neuron. For DC measurement, the Agilent Semiconductor Parameter Analyzer was deployed to sweep diode voltage $V_{DS}$ from −2 V to 2 V, and diode current $I_{DS}$ was recorded at a constant humidity of 58% and temperature of 27 °C. In the second measurement, the Agilent Semiconductor Parameter Analyzer was utilized to drive the ground potential, and two Keithley 3390 waveform generators generated input step pulses for $V_{IN1}$/$V_{IN2}$, alternating between −2 and 2 V and shifted in time by 90°. Output signals $V_{x}$ and $V_{out}$ were measured at a constant humidity level of 50% and a temperature of 27 °C in parallel with using the digital oscilloscope.

### 2.3. Bending Tests

We performed the bending tests on the fully printed devices on a cylinder with 4.5 mm radius for over 200 cycles. Transfer characteristics of the EGT were recorded before and after bending. Scanning electron micrographs and energy-dispersive X-ray mapping were performed directly on the bent substrates (Appendix A).

## 3. Results

### 3.1. Fabrication of EGT on Polyimide

In order to realise fully printed neurons on a flexible substrate, the processing temperature of the functional materials must be optimized to suit the substrate. The glass transition temperature (T_g_) of polyimide occurs between 360 and 400 °C [42]. Therefore, printed films are processed at lower temperatures. While earlier works used temperatures as high as 400 °C to obtain In${}_{2}$O${}_{3}$ films from halide precursors [43], we demonstrate that these films can be obtained from nitrate precursors at lower temperatures owing to the low decomposition temperature of $\mathrm{In}{\left(NO_{3}\right)}_{3}$ and the low crystallization temperature (290 °C) of In${}_{2}$O${}_{3}$ [44]. The quality of indium oxide obtained from an aqueous $\mathrm{In}{\left(NO_{3}\right)}_{3}$ solution subjected to a heat treatment was comparable whether annealed to 300 or 400 °C. This was confirmed by X-ray diffraction (Figure 3a) on the same film, annealed first to 300 °C and then to 400 °C. As expected from the literature [45], crystallite size increased with temperature from 4.6 nm at 300 °C to 10.45 nm at 400 °C, as calculated per the Scherrer formula [46] for the highest intensity peak at (222). Typical values for indium oxide prepared by pulsed laser deposition were 19.8 nm [47], but for solution-processed polycrystalline indium oxide films, 4.6 nm is a very competitive value.

For this study, the most important conclusion from the XRD pattern was that characteristic In${}_{2}$O${}_{3}$ peaks were equally observed at the lower-temperature treatment with no undesirable peaks. Subsequently, 300 °C was chosen as the maximal processing temperature in this work.

Passive graphene structures, i.e., electrodes and resistors, were annealed along with an In${}_{2}$O${}_{3}$ precursor. After heating to 300 °C, a graphene film of 300 nm thickness on a glass substrate showed sheet resistance of 500 Ω [40]. The inset in Figure 3b shows an optical image of the film that was also used for SEM to determine the average flake size of 200 nm (image analysis in ImageJ). In order to realize different resistances as required for the conductors or resistors, the thickness of the printed graphene film was varied to ensure that the printed area footprint was as small as possible. Table 1 shows that the graphene film sheet resistance decreased with each additional printed layer for a variation of temperatures up to 300 °C. Figure 3b validates the inverse relation of the graphene sheet resistance to its thickness. In conventional inks, e.g., silver ink [48], the resistance can be changed by either increasing the length by a very large measure or decreasing the thickness to below the possible minimum for printing. For graphene, however, we can go the opposite way and add layers to tune the resistance to our requirements without increasing the printing area.

The good printability of both the oxide precursor and the graphene ink was achieved by preconditioning the substrate (for details, see Section 2). The contact angles of the printed films were 78° for graphene ink (Appendix A) and 38° for aqueous $\mathrm{In}{\left(NO_{3}\right)}_{3}$ (Appendix A), with an angle < 90° indicating that good wettability was ensured. A processing temperature below the glass transition temperature of the PI substrate helps in avoiding crack formation in In${}_{2}$O${}_{3}$. The morphology of the processed In${}_{2}$O${}_{3}$ was probed using the atomic force micrograph seen in Appendix A at different areas on the In${}_{2}$O${}_{3}$ surface. This showed a homogeneous morphology without crack formation, which is crucial for electrical performance. A scanning electron micrograph (SEM, Figure 2c) of the interface of graphene-In${}_{2}$O${}_{3}$ indicated the contrast in the roughness of the two materials, and showed good contact without any delamination. Graphene film roughness (average of 40 nm) was clearly several times greater than that of In${}_{2}$O${}_{3}$, which had an average roughness of 6 nm. Detailed information on the fabrication steps is provided in Section 2.

Since the In${}_{2}$O${}_{3}$ semiconductor dimensions matched those in our previous work [39], the tensile strain that could be applied on the fully printed devices without any damage was 2%, which amounts to a 4.5 mm extension. While the PI substrate, graphene film, and CSPE would allow for larger tensile strains, even as high as 100% [49,50] thanks to the presence of polymer inside them (Section 2), the oxide semiconductor is brittle in comparison. Therefore, the bending radius was limited by the oxide semiconductor. However, given the small size of In${}_{2}$O${}_{3}$ in comparison with that of polymer-containing components, it is possible to bend the substrate at higher tensile strain. On the other hand, since a 4.5 mm bending radius is sufficient for most flexible applications [51], this value was the minimal bending radius for the printed neurons. To test bending stability, we examined the EGT characteristics before and after 200 bending cycles to 2% strain (Figure 4). The difference between the transfer curves before and after bending was negligible. The threshold voltage increased very slightly, while the on current remained constant. No cracks occured in the semiconductor, as confirmed by using the energy-dispersive X-ray mapping of elements directly on the bent device (Appendix A).

With regard to the longevity of the devices, performance depends on the electrolyte and the drying effects. We tested the transfer characteristics of EGTs over four weeks, during which performance was stable. The transfer characteristics of the EGT initially and after four weeks (Appendix A) are shown in the Appendix A. Thus, the device achieved consistent performance for several bending cycles over several weeks.

### 3.2. Printed Flexible 2-Input Neural Network

The fabrication steps of the printed EGT-based two-input neuron are depicted in Figure 5. As substrate material, flexible polyimide (PI, Kapton^®^ HN, Dupont, Wilmington, DE, USA) was deployed. To counter any differences in the thermal expansion of substrate and films to be deposited, the PI was initially annealed at 400 °C, and then rinsed with a 1:1 mixture of acetone and isopropanol [39]. Source, drain, and gate electrodes, contact pads, and $R_{out}$, $R_{1}$, $R_{2}$ and $R_{3}$ resistances were printed according to the desired resistance values. After printing the $\mathrm{In}{\left(NO_{3}\right)}_{3}$ precursor, the whole substrate was annealed at 300 °C for two hours (ramp rate 1 °C/min) to form the In${}_{2}$O${}_{3}$ semiconductor attached to the source or drain. Once the passive structures had been printed and ready, the CSPE electrolyte was printed across the gate and the $\mathrm{In}{\left(NO_{3}\right)}_{3}$ semiconductor to construct a transistor-based diode.

The analog circuit design of the fully printed two-input neuron is illustrated in Figure 6a. It consisted of the MAC operation combined with a printed piecewise linear unit (pPLU) that included the diode. An optical microscope image of the fabricated circuit on flexible polyimide foil is shown in Figure 6b; voltage signals (colored pads) and printed materials are annotated. The design parameters of the printed hardware prototype are depicted in Table 2.

The fabricated flexible neuron realized the two fundamental building blocks to implement artificial neural networks: the multiply–accumulate (MAC) and nonlinear-activation functions [29]. These operations can be expressed as follows:(1)$$\begin{array}{cc} z(x_{i}) & =\sigma (\sum_{i}x_{i}\;w_{i}), \end{array}$$
where summation $\sum_{i}x_{i}\;w_{i}$ is the MAC operation, and σ(·) is the (preferably nonlinear) activation function. To realize these functions, analog designs were chosen. To this end, quantities $x_{i}$ and *z* were encoded as voltages ($V_{IN_{i}}$ and $V_{out}$), and ANN weights $w_{i}$ and the MAC operation were realized with resistors [33]. The applied input voltages were converted into proportional currents to resistor conductance according to Ohm’s law (multiplication). These currents were then summed up at the T-junction ($V_{x}$ in Figure 6a) following Kirchhoff’s rule (addition). As resistors are linear circuit elements, the output voltage of the resistor can be computed by solving the linear problem of a resistor network, where resistors are known, and the output voltage is the unknown variable [34].

Output voltage $V_{x}$ of the MAC operation for our two-input neuron was obtained from
(2)$$\begin{array}{cc} V_{x} & =\sum_{i}V_{IN_{i}}\;w_{i}=V_{IN1}\;w_{1}+V_{IN2}\;w_{2}, \end{array}$$
with
(3)$$w_{i}=\frac{\frac{1}{R_{i}}}{\left(\sum_{j}\frac{1}{R_{j}}\right)}=\frac{\frac{1}{R_{i}}}{\left(\frac{1}{R_{1}}+\frac{1}{R_{2}}\right)},$$
where i,j∈{1,…,N} denote the index of the *N* input resistors $R_{i},R_{j}$.

Since activation functions for ANNs must provide nonlinear behavior between its inputs and output, a suitable candidate for implementation is a diode-based circuit. Hence, activation function ($V_{out}=\sigma \left(V_{x}\right)$) in this work was realized with an analog diode-based circuit ($T_{1}$ and $R_{H}$) in combination with a pull-down resistor $R_{out}$ (see Figure 6a) to convert the diode current into output voltage. This allows for cascading several neurons to obtain multiple ANN layers, as each neuron inputs and outputs voltages. As a result, total neuron output *z* could be obtained by computing the voltage drop across $R_{out}$ caused by diode current $I_{DS}$. Using Kirchhoff’s rule, the following equation is obtained: (4)$$\begin{array}{c} V_{out}=\sigma \left(V_{x}\right)=I_{DS}\left(V_{x}-V_{out}\right)\cdot R_{out}, \end{array}$$
where $I_{DS}\left(V_{x}-V_{out}\right)$ is a function of voltage drop $V_{x}-V_{out}$ across the diode. In contrast to Equation (Equation 2), Equation (Equation 4) must be numerically solved.

## 4. Discussion

### 4.1. Measurement Results

For the electrical characterization of the printed two-input neuron, we first measured the DC behavior of the printed EGT-based diode, which is part of the pPLU activation function and allows for extracting the transfer function of the pPLU corresponding to Equation (Equation 4). The DC measurement results of the EGT-based diode in Figure 7a clearly show that the diode had an input voltage regime with a high forward current ($I_{DS}$) and a low current in the reverse direction. The diode’s forward threshold voltage was about 750 mV, which was suitable to observe a nonlinear response of the diode within the operating voltage. Figure 7a also shows the computed voltage of the pPLU, which validated the nonlinear behavior of this analog activation function circuit. As the resulting resistance was already high enough to obtain a suitable output voltage swing for the pPLU, only one-fourth of the length of the printed graphene-based resistor was used for the pPLU and neuronal measurements (measurement needle for GND came into contact in the middle of the first meander of $R_{out}$), leading to $R_{out}=3.75\;$MΩ (instead of 4×3.75MΩ=15MΩ). Such high $R_{out}$ resistances do not affect the MAC output, which had resistances $R_{i}$ below 1 KΩ, and currents in the MAC circuit were much higher than those in the pPLU.

In the second measurement step, we analyzed the MAC circuit and the overall two-input neuronal behavior. Figure 7b shows the transient measurements of the printed neuron. Output $V_{x}$ of the MAC operation was the sum of inputs $V_{IN1}$ and $V_{IN2}$, as expected, i.e., when the inputs were complementary, the MAC output $V_{x}$ was close to 0 V, as resistors $R_{1}$ and $R_{2}$ were of equal size, and the resulting weights were $w_{1}=w_{2}=0.5$, respectively. When both inputs were pulled up to 2 V or −2 V, MAC output voltage $V_{x}$ was likewise. Second, the nonlinear response of the pPLU could also be obtained with the measured waveforms. When pPLU input $V_{x}$ was negative, output $V_{out}$ was suppressed; inversely, for positive input voltages, $V_{out}$ was pulled up to a higher voltage level. In other words, the pPLU response was nonlinear with respect to the MAC output, as desired. In summary, we successfully demonstrated the overall function of the fully printed neuron prototype, and validated that both fundamental neuron operations operated correctly.

### 4.2. Outlook

From the transient measurements (Figure 7), an input–output delay of 1 s was obtained with an average power consumption of 37.7 mW, the calculation of which is shown in Appendix A. The consumed area of the hardware prototypewas as small as 16.25 mm^2^. However, a substantial reduction in power consumption could be achieved by choosing smaller resistances of $R_{i}$, and with a smaller area of printed resistors and contact pads. This is feasible, as a predefined set of ANN weights can be realized by differently sized resistors, e.g., choosing a common scale factor for all resistors $R_{i}$ in Equation (Equation 3) [34]. Predefining resistors through the training model also allows for the usage of printed resistors, which are positive weights that cannot be dynamically tuned. We, therefore, assumed that our neuron was trained beforehand in software and fabricated accordingly, which basically rendered our ANN as a “post-trained” network that could perform near-sensor processing. Following this approach, our NN is not run-time-adaptive, but targets applications that require low-cost fabrication on flexible substrates. In order to tune the weights, conductive materials such as Pedot:PSS can be printed on top, and this is being investigated. Furthermore, the number of neuronal inputs is usually not limited to two; by adding additional resistors $R_{i}$ to the MAC, inputs can be scaled up in accordance to taking into account the maximal voltage headroom defined by the input signal and power supply.

As the interconnects of the circuit, such as measurement pads for drain, source, gate, GND or $R_{out}$, parasitic resistances are introduced to the circuit of the neuronal hardware prototype; their area and consequently their conductance were enlarged compared to passive components $R_{1}$, $R_{2}$ and $R_{out}$. Although input contact pads $V_{IN1}$ and $V_{IN2}$ influenced the effective resistance of input resistors $R_{1}$ and $R_{2}$ when measuring the hardware prototype, they did not impact ANN weights. Only the ratio of the effective resistance determines the ANN weights, not their absolute value; see Equation (Equation 3). In our experiment, both weights were set to 0.5, and input pads and resistors $R_{1}$ and $R_{2}$ were of equal size. However, when a neuron is fabricated as part of a commercial product, no measurement pads are required, and only the output impedance of the connected input device (e.g., a sensor) must be lower than neuron input resistors $R_{i}$.

## 5. Conclusions

We successfully fabricated a two-input fully printed neuron on flexible polyimide using drop-on-demand additive-printing techniques. To the best of our knowledge, this is the first time that such a demonstrator is reported that integrates digital printing techniques with flexible substrates in this field of application. The software-pretrained neuron is based on graphene ink for the conductive tracks, resistors, and EGT source, gate, and drain electrodes. The resistance of the printed graphene film was modulated by changing its thickness. The heating process, simultaneously annealing both graphene ink and In${}_{2}$O${}_{3}$, was optimized, such that there was no crack formation in In${}_{2}$O${}_{3}$. The transistor-based diode used electrolyte gating; thereby, the two-input neuron operated at only 2 V, rendering it a suitable candidate for self-powered portable printed-computing systems. The printed neuron had propagation delay of 1 s, power consumption of 37.7 mW, and area utilization of 16.25 mm^2^, paving the road for upscaling fully printed two-input neurons.

The presented NCS concept can be further improved by extending currently used ANN building blocks with circuits for negative weight operation. Another extension would be to use a nonlinear activation function. Our future work will implement these changes to fabricate an extended neuronal concept while using the fully printed approach on flexible substrate.

## Figures and Tables

**Figure 1 sensors-22-04000-f001:**
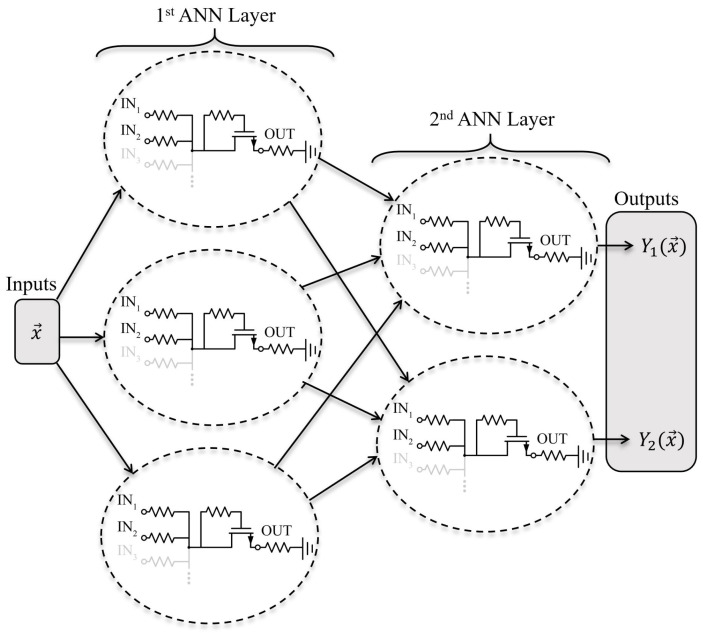
Topology of printed ANN with the proposed neuron design, two layers, and a topology of 3×2.

**Figure 2 sensors-22-04000-f002:**
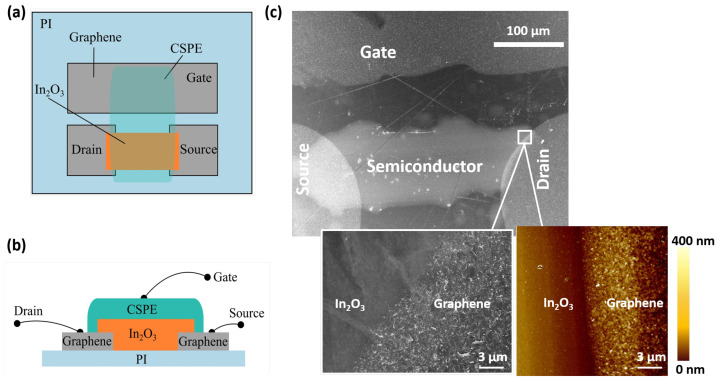
(**a**) Top and (**b**) side views of the printed transistor. (**c**) Top-view SEM image of the transistor structure. Inset shows SEM and AFM micrographs of the interface between graphene film and In${}_{2}$O${}_{3}$. Average graphene flake size was about 200 nm. Atomic force micrography shows that the graphene ink had higher roughness of more than 200 nm as compared to that of the indium oxide film, which had a roughness of 6 nm.

**Figure 3 sensors-22-04000-f003:**
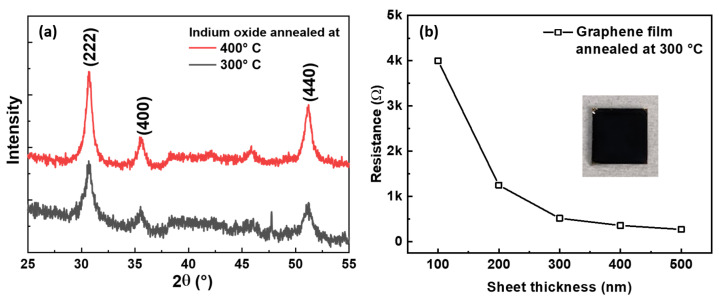
(**a**) XRD diffraction pattern of indium oxide obtained from aqueous nitrate precursors. Films obtained at 300 °C had characteristic peaks of indium oxide and did not show any unwanted peaks. (**b**) Variation in graphene sheet resistance with layer thickness showed inverse proportionality. Inset shows a photograph of a printed graphene film on 5 × 5 mm${}^{2}$ glass substrate annealed at 300 °C as used for resistance dependance measurement.

**Figure 4 sensors-22-04000-f004:**
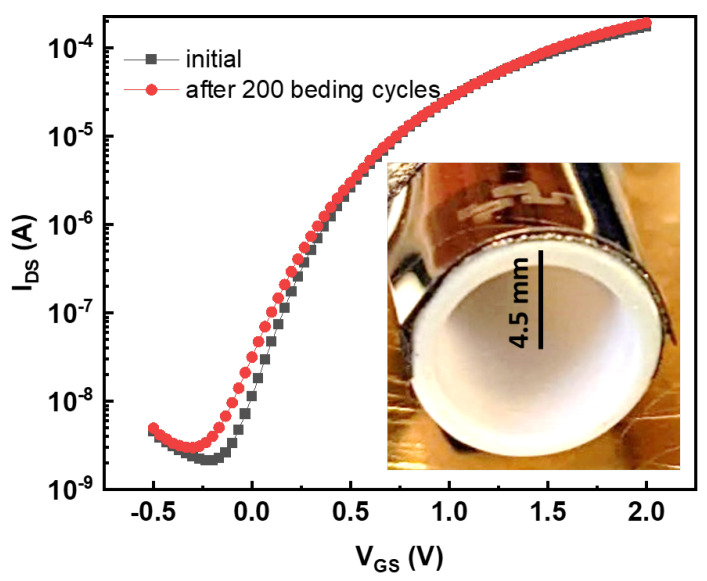
Transfer characteristics of electrolyte gated transistor before and after 200 bending cycles showed merely negligible difference between curves. Inset shows devices on the polyimide substrate bent to a 4.5 mm radius.

**Figure 5 sensors-22-04000-f005:**
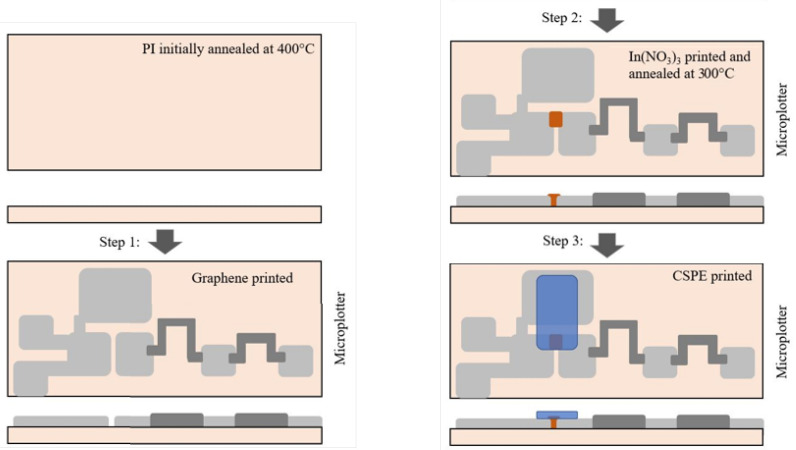
Neuron fabrication steps using the printing equipment described in Section 2. For each step, the top and side views of the printed structure are provided. Polyimide was preheated to 400 °C. In Step 1, graphene of varied thickness was printed, reflected in magnitude of resistance postannealing. In Step 2, aqueous indium nitrate was printed, and the substrate was annealed at 300 °C. In Step 3, CSPE was printed across the gate and In${}_{2}$O${}_{3}$.

**Figure 6 sensors-22-04000-f006:**
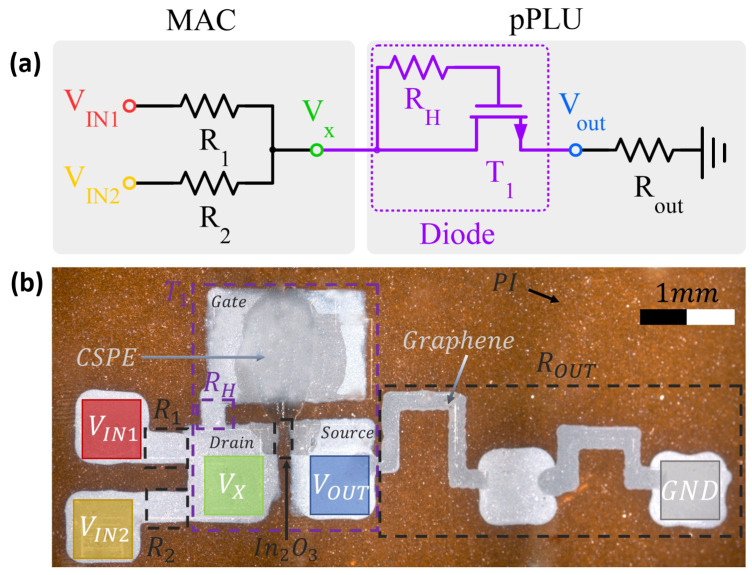
Hardware prototype of fully printed two-input neuron: (**a**) Circuit schematic; (**b**) annotated optical micrograph of the printed two-input neuron on flexible polyimide film. The image was digitally postprocessed and stitched together from multiple microscopy photos.

**Figure 7 sensors-22-04000-f007:**
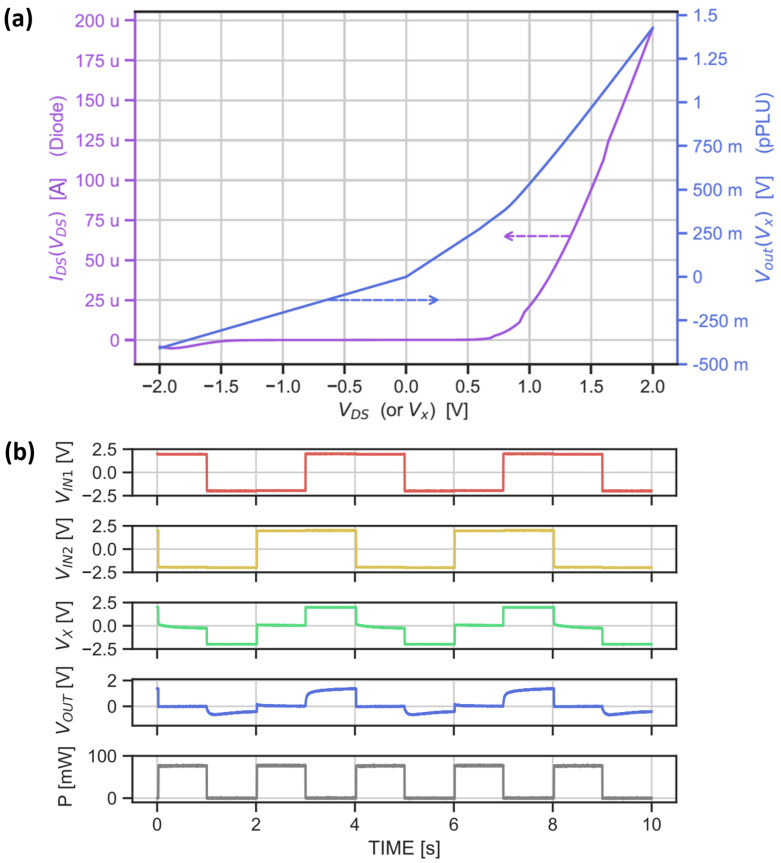
(**a**) DC measurement of EGT-based diode ($I_{DS}\left(V_{DS}\right)$) and activation function ($V_{out}\left(V_{x}\right)$). (**b**) Transient measurements of the printed neuron.

**Table 1 sensors-22-04000-t001:** Sheet resistance for one to five layers of graphene ink printed and annealed at different temperatures. Each layer was 100 nm thick.

	100 °C	200 °C	250 °C	300 °C
1 layer	6 GΩ	160 MΩ	12 kΩ	4000 Ω
2 layers	5 GΩ	135 MΩ	8 kΩ	1250 Ω
3 layers	5 GΩ	114 MΩ	7 kΩ	520 Ω
4 layers	3 GΩ	96 MΩ	2 kΩ	360 Ω
5 layers	1 GΩ	55 MΩ	1 kΩ	275 Ω

**Table 2 sensors-22-04000-t002:** Design parameters of the printed neuron hardware prototype.

R${}_{1}$	R${}_{2}$	R${}_{H}$	R${}_{\mathrm{out}}$	T${}_{1}$-Width	T${}_{1}$-Length	Diode V${}_{\mathrm{th}}$
250 Ω	250 Ω	300 Ω	3.75 MΩ	400 μm	140 μm	750 mV

## Data Availability

Data supporting reported results can be accessed from the authors upon request.

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
