# Peer review of "Artificial Neurons on Flexible Substrates: A Fully Printed Approach for Neuromorphic Sensing"

_sensors, 2022, doi:10.3390/s22114000_

Round 1

Reviewer 1 Report

In this paper, novel demonstration for neuromorphic sensing is described. The content is interesting for many readers. Although bendability of the presented results are rather poor, it should be published as it is.

Although similar topics are reported in the review paper by H. Ling [1], this paper demonstrates the fully printed tow-input neuron circuit on flexible substrate for the first time. This point is the originality of this paper.
The review paper[1] should be referred and discussed in the introduction section.

[1] Haifeng Ling,Dimitrios A. Koutsouras, Setareh Kazemzadeh, Yoeri van de Burgt, Feng Yan, and Paschalis Gkoupidenis,
Electrolyte-gated transistors for synaptic electronics, neuromorphic computing, and adaptable biointerfacing
Appl. Phys. Rev. 7, 011307 (2020)

In Fig.1(c), the top-view image of the transistor structure is quite poor.
It would be much more persuasive for the readers that the image is clearer.d

Reviewer 2 Report

Dear Authors,

in your manuscript, you report a fully printed neuromorphic computing system with an in-depth study of the morphology and electrical properties of printed graphene and indium oxide. Although the work is extremely interesting, in my opinion, it does not fit very well in this special issue/journal. This is mainly because it lacks a sensing application.

Moreover, I think the concept of neuromorphic computing is not really known by most of the readers of the journal and it should be introduced better. It has been difficult for me to follow the dissertation and I had to look for external sources of information, not suggested in the text.

Another issue I found in the manuscript is that the theoretical part should actually be presented before the experimental results in my opinion (pay also attention to where you define/cite the acronyms in the paper).

I would recommend you submitting your work to another journal complying more with the topic of your paper. However, if you prefer to keep this journal I think that a sensing application should be presented, even with just a proof of concept experiment.

Thanks for your work and good luck,

Best regards.

Reviewer 3 Report

Comments:

In this paper, the authors reported a fully printed artificial neuromorphic circuit on flexible PI substrate. The resistance of the printed graphene film can be modulated by changing its thickness. Some valuable conclusions were obtained. This work is interesting, but some results don't have convincing explanations in this article. My comments are shown below:

  1. Some related important references on printing of sensors should be cited (ACS Applied Materials & Interfaces, 2020, 12(25): 28669-28680), which can increase the credibility.
  2. The sizes of the printed transistor structure should be given in the paper.
  3. Figure S3 should be moved to the main text to show the interface between graphene and In2O3.
  4. I think the printed devices is deformable due to the presence of polymer. Why the authors said that the printed devices can be stretched to 2% without ang damage? The reason should be given.
  5. Figure S5 shows the transfer characteristics of the device after four weeks. Is the device under bending loading for four weeks? If not, the long-term cyclic bending testing should be given to indicate the stability.

Reviewer 4 Report

Have a look my comments

Round 2

Reviewer 2 Report

Dear Authors,

honestly, I am not completely satisfied by your reply and the lacking of a sensing application in the paper. This journal in my opinion should indeed point in that direction. However, I recognise the interest of the topic and results you are presenting for the readership of the journal and I decided to accept your manuscript after some minor but important revisions. Let me list them here:

1-Please, explain better in the introduction the value of neuromorphic computing. Consider that most of the readers probably have never heard about it so try to be more explicative. As an example, I found the introduction by Weller et al. 2021 scientific reports accompanying better the reader through the topic.

2-In your paper you still talk about MAC and pPLU (row 223) much before defining them (rows 229 and 257) and this is confusing.

3-Clarify the issues related to non-negative weighting and at least propose methods to allow it in a future all-printed system.

4-Set the difference between your system and the one proposed by Weller et al. in Scientific reports. What's the novelty in your approach?

Thanks for your work and best regards,
